# LC-UV and UPLC-MS/MS Methods for Analytical Study on Degradation of Three Antihistaminic Drugs, Ketotifen, Epinastine and Emedastine: Percentage Degradation, Degradation Kinetics and Degradation Pathways at Different pH

**Anna Gumieniczek** [1,*] , **Izabela Kozak** [1], **Paweł Żmudzki** [2] **and Urszula Hubicka** [3]

1 Department of Medicinal Chemistry, Medical University of Lublin, Jaczewskiego 4, 20-090 Lublin, Poland; izakozak7@gmail.com
2 Department of Medicinal Chemistry, Jagiellonian University, Collegium Medicum, Medyczna 9, 30-688 Cracow, Poland; zmudzki.p@gmail.com
3 Department of Inorganic and Analytical Chemistry, Jagiellonian University, Collegium Medicum, Medyczna 9, 30-688 Cracow, Poland; urszula.hubicka@uj.edu.pl
* Correspondence: anna.gumieniczek@umlub.pl; Tel.: +48-81-4487380; Fax: +48-81-4487381

**Abstract:** Evaluation of pH-dependent reactivity of drugs is an essential component in the pharmaceutical industry. Thus, the stability of three antihistaminic drugs, i.e., ketotifen, epinastine and emedastine, was tested, in solutions of five pH values, i.e., 1.0, 3.0, 7.0, 10.0 and 13.0, at high temperature (70 °C). LC-UV isocratic methods were developed to estimate percentage degradation as well as the kinetics of degradation. Generally, epinastine was shown to be the most stable compound with degradation below 14%. Emedastine was labile in all pH conditions, with degradation in the range 29.26–51.88%. Ketotifen was moderately stable at pH 1–7 (degradation $\leq$ 14.04%). However, at pH $\geq$ 10, its degradation exceeded 30%. The kinetics of degradation of ketotifen, epinastine and emedastine was shown as a pseudo-first-order reaction with the rate constants in the range $10^{-4}$–$10^{-3}$ min$^{-1}$. Finally, the UPLC-MS/MS method was applied to identify the main degradants and suggest degradation pathways. Degradation of ketotifen proceeded with oxidation and demethylation in the piperidine ring of the molecule. As far as epinastine was concerned, opening of the imidazole ring with formation of the amide group was observed. Unfortunately, no degradation products for emedastine were detected. The present results complete the literary data and may be important for both manufacturing of these drugs and their administration to patients.

**Keywords:** LC-UV and UPLC-MS/MS methods; degradation in solutions; pH and high temperature; new degradation products; ketotifen; epinastine and emedastine

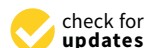



## 1. Introduction

Susceptibility of drugs to degradation may fluctuate in connection with their chemical structures and reactivity as well as with types of formulations. For many drugs and dosage forms, their sensitivity can lead to chemical or physico-chemical changes [1]. Among others, different pH conditions can affect the drug stability that may finally cause some undesired reactions in patients. Thus, evaluation of the pH-induced reactivity of drugs is an essential component in the pharmaceutical industry [2].

Ketotifen (KETO), epinastine (EPI) and emedastine (EME) belong to the drugs that competitively inhibit the action of histamine on tissues containing H$_1$ receptors and are commonly used for the treatment of different forms of allergies. The H$_1$ antagonists are divided into two broad groups, i.e., the first generation known as "classical" antihistamines and the second generation or "no sedating" antihistamines. Depending on their chemical structures, the H$_1$ receptor antagonists can be divided into ethanolamines, ethylenedi-amines, propylamines, piperazines, piperidines and tricyclic antagonists as well as active

optical isomers of the parent drugs or their metabolites [3]. Generally, the structures of the $H_1$ receptor antagonists present a diaryl substitution pattern and contain an amine function, both of which are essential for the $H_1$ receptor affinity. The amino moiety could also be important to obtain the salts of basic drugs for manufacturing their sufficiently stable dosage forms [4]. KETO, EPI and EME belong to the second-generation agents having little affinity for muscarinic, adrenergic or serotoninergic receptors and therefore displaying lower side effects. EME is a competitive blocker of the $H_1$ receptor, whereas KETO and EPI are dual action agents combining both antihistaminic and mast cell-stabilizing properties [5]. Their chemical structures are shown in Figure 1.

KETO                    EPI                    EME

**Figure 1.** Chemical structures of ketotifen (KETO), epinastine (EPI) and emedastine (EME).

Only two HPLC methods including a stability study have been reported in the literature for KETO [6,7]. As far as mass spectrometry is concerned, two LC-MS methods were elaborated for different pharmacokinetics purposes [8,9]. The literature on EPI showed three stability-indicating HPLC methods for its determination in pharmaceuticals [10–12], where its stability was examined as a part of the validation procedure. At the same time, LC-MS methods from the literature were only used for quantifying EPI in human plasma [13,14]. As far as EME is concerned, a literature survey revealed one stability-indicating TLC method which enables estimation of chemical degradation of EME [15]. As far as LC-MS methods are concerned, only two reports concerning determination of EME or its metabolites in plasma were reported [16,17].

The above data show that there is very little information on the chemical stability of KETO, EPI and EME under various storage and stress conditions. What is more, no papers on the identification of their decomposition products using LC-MS methods were found in the literature. On the other hand, previous experiments from our laboratory showed the sensitivity of KETO, EPI and EME to UV/VIS light in the range 300–800 nm and their pH-dependent photodegradation [18]. Considering the importance of these frequently prescribed drugs and the lack of reports concerning their stability, the primary goal of the present experiment was to explore their sensitivity to different pH conditions at high temperature. The specific goals were to elucidate the degradation kinetics and degradation pathways and identify degradation products of KETO, EPI and EME, using LC-UV and UHPLC-MS/MS methods. High temperature (70 °C), the buffers of pH 3.0, 7.0 and 10.0 and 0.1 M HCl (pH 1.0) and 0.1 M NaOH (pH 13.0) were used as degradation solutions, since the degradation processes can be dependent on the ionized forms of the molecules.

## 2. Materials and Methods

### 2.1. Apparatus

A 515 pump with a Rheodyne injector and a 20 µL loop together with an UV 2487 DAD detector controlled by the Empower software version 3.0 from Waters Corporation (Elstree, Herts, England) were used for our LC-UV method. A Waters Acquity UPLC apparatus with a Waters eλ PDA detector and a Waters TQD mass spectrometer (an ESI tandem quadrupole) from Waters Corporation (Milford, MA, USA) were used for

our UPLC-MS/MS analysis. A HI9024C pH meter from Hanna Instruments (Ronchi di Villafranca Padovana, Italy) and a thermostated water bath from WSL (Warsaw, Poland) were also employed.

*2.2. Materials*

Ketotifen hydrogen fumarate (KETO), epinastine hydrochloride (EPI), emedastine difumarate (EME), chlorprotixene hydrochloride and todralazine hydrochloride (internal standards for LC-UV methods) were purchased from Sigma-Aldrich (St. Louis, MO, USA). Acetonitrile, methanol and formic acid for LC analysis from Merck (Darmstadt, Germany) were used in our LC-UV methods. Formic acid, ammonium formate, acetonitrile and water for LC-MS analysis from J.T. Baker (Center Valley, PA, USA) were used in our UPLC-MS/MS method. All other reagents used for preparing buffers and degradation solutions were of analytical grade and were purchased from POCh (Gliwice, Poland). Zabak® drops (0.25 mg/mL of KETO) from Théa Pharma (Schaffhausen, Switzerland), Relestat® drops (0.5 mg/mL of EPI) from Allergan (Westport, Ireland) and Emadine® drops (0.5 mg/mL of emedastine) from Novartis Europharm Ltd. (London, United Kingdom) were purchased from the local pharmaceutical market.

Prior to creating acidic, neutral or alkaline samples for degradation studies, three buffer solutions were made at a constant ionic strength of 1 M adjusted with 4 M sodium chloride. The pH ranged from an acidic range of 3.0 (acetate buffer) through neutral conditions of 7.0 (phosphate buffer) to basic conditions at 10.0 (borate buffer). In addition, for extreme pH conditions, 0.1 M HCl and 0.1 M NaOH were used. Meanwhile, phosphate buffer of pH 2.5 and acetate buffer of pH 4.8 were prepared for our LC-UV methods. All buffer solutions were freshly prepared, and their pH was measured at a room temperature of $23 \pm 2$ °C.

*2.3. LC-UV Methods*

2.3.1. Chromatographic Conditions

All LC analyses were carried out at room temperature $23 \pm 2$ °C using isocratic elution. A LiChrospher®100RP-18 column from Merck ($125 \times 4.0$ mm, 5 μm) and a mobile phase consisting of acetonitrile, methanol and acetate buffer of pH 4.8 (30:40:30, *v/v*) containing 0.25% formic acid with the flow rate of 2.0 mL/min were used for determination of KETO. Detection was performed spectrophotometrically at 296 nm. For EPI and EME assays, the same LiChrospher®100CN column from Merck ($125 \times 4.0$ mm, 5 μm) was applied. EPI was determined using the mixture of acetonitrile, methanol and acetate buffer of pH 4.8 (45:35:20, *v/v*), with the flow rate of 1.5 mL/min, whereas detection was conducted at 240 nm. The mobile phase for EME was the mixture of methanol and phosphate buffer of pH 2.5 (40:60, *v/v*), with the flow rate of 1 mL/min, while the UV detector was set at 280 nm. Chlorprotixene was used as an internal standard for KETO and EME assays, while todralazine was the optimal internal standard for EPI. The methods were validated before application to the assays of KETO, EPI and EME in the stressed samples.

2.3.2. Selectivity

Selectivity of the methods was examined by determination of KETO, EPI or EME in the presence of internal standards, degradation products or excipients from respective ocular drops.

2.3.3. Stock Solutions

Stock solutions of KETO, EPI, EME and internal standards (chlorprotixene and todralazine) were prepared in methanol at a concentration of 1 mg/mL or 10 mg/mL. They were stored in the dark at 4 °C and were found stable for at least 3–4 weeks.

### 2.3.4. System Suitability

Six solutions of KETO, EPI or EME were prepared by dispensing 1.0 mL volumes from the stock solutions to 10 mL volumetric flasks, to reach the concentrations of 100 μg/mL. The proper volume of the internal standard solution was added to each flask. After adjusting with methanol to the mark, 20 μL volumes were injected onto the column.

### 2.3.5. Calibration of the Methods

Working solutions of KETO, EPI and EME were prepared by pipetting 0.1–1.0 mL volumes of respective stock solutions to 10 mL volumetric flasks, to obtain the concentration range 10–100 μg/mL. For KETO samples, 0.5 mL volumes of chlorprotixene solution of 10 mg/mL were added to each flask. For EPI samples, 0.2 mL volumes of todralazine solution of 1 mg/mL were added to each calibration solution. For EME samples, 0.6 mL volumes of chlorprotixene solution of 1 mg/mL were added to each flask. All samples were filled with methanol to the mark and five injections were conducted onto the column, for each drug and each concentration. The calibration curve was constructed by plotting the peak area ratios (peak area of the drug versus peak area of the internal standard) against the corresponding concentrations of the respective drug, using the least squares method. The SD values of the intercepts and the slopes of the respective regression lines were used for calculation of the limits of detection (LOD) and limits of quantification (LOQ) for KETO, EPI and EME.

### 2.3.6. Precision

Precision of the methods was verified by replicate injecting the solutions of KETO, EPI and EME at three concentrations (low, medium and high). The solutions of KETO at concentrations of 30 μg/mL, 50 μg/mL and 70 μg/mL were used. As far as EME and EPI were concerned, the solutions at concentrations of 15 μg/mL, 50 μg/mL and 90 μg/mL were injected. The solutions were injected onto the column three times on the same day, and then on the next three days.

### 2.3.7. Accuracy

Accuracy of the methods was estimated by determining KETO, EPI and EME in commercially available ocular drops and comparing the determined amounts to the nominal values. The six volumes of 1.0 mL of Zabak® drops or 0.5 mL of Relestat® and Emadine® drops were transferred to 5 mL volumetric flasks, and respective aliquots of the stock solutions of the internal standards were added. The solutions were diluted with methanol to the mark and analyzed by the LC-UV methods described above.

### 2.4. Degradation at Different pH and High Temperature

From the stock solution of KETO, EPI or EME (1 mg/mL), the volumes of 1 mL were pipetted to small glass tubes (Medlab, Raszyn, Poland). To each tube, 1 mL volumes of the appropriate degradation solutions (0.1 M HCl, buffers of 3.0, 7.0, 10.0 and 0.1 M NaOH) were added. The tubes were tightly closed with stoppers and placed in a water bath set at 70 °C. The samples were removed from the bath after subsequently 30, 60, 90, 120, 150, 180, 210, 240, 270 and 300 min. They were immediately cooled, neutralized if necessary, diluted with methanol to cover the linearity range, mixed with the respective volumes of the internal standard solutions and analyzed by the LC-UV methods. The measurements were repeated three times for each drug and each time point of degradation. Finally, the stressed samples of KETO, EPI and EME were analyzed by our UPLC-MS/MS method.

### 2.4.1. Kinetics of Degradation

The concentrations of non-degraded KETO, EPI and EME remaining after each time point of degradation were calculated using respective calibration equations. Then, these concentrations or their logarithms were plotted against time of degradation, to obtain the equations $y = ax + b$ and the determination coefficients $r^2$. Finally, respective kinetic

parameters, i.e., a constant degradation rate (k) and degradation time of 50% substance ($t_{50}$), were calculated for each drug.

### 2.4.2. Degradation Pathways Using UPLC-MS/MS Method

The Acquity UPLC BEH C18 column (100 × 2.1 mm, 1.7 μm) and the Acquity UPLC BEH C18 VanGuard precolumn (5 × 2.1 mm, 1.7 μm) from Waters Corporation (Milford, MA, USA) were used for chromatographic separation at 40 °C. The gradient elution with eluent A decreasing from 95% to 0% over 10 min at a flow rate of 0.3 mL/min was applied. Eluent A was the mixture of water and formic acid (0.1%, *v/v*), while eluent B was the mixture of acetonitrile and formic acid (0.1%, *v/v*). After recording the chromatograms, spectra were analyzed in the range 200–700 nm with a 1.2 nm resolution and sampling rate of 20 points/s.

The TQD mass spectrometer worked in positive ESI mode using the following settings: source temperature 150 °C, desolvation temperature 350 °C, desolvation gas flow rate 600 L/h, cone gas flow rate100 L/h, capillary potential 3.00 kV and cone potential 30 V. Nitrogen was used as both the nebulizing and drying gases. The scan mode from 50 to 1000 *m/z* at 0.5 s intervals was applied. Collision-activated dissociation (CAD) analyses were carried out with the energy of 50 eV. Consequently, the ion spectra were obtained by scanning from 50 to 500 *m/z*. The MassLynx software version 4.1 from Waters Corporation was used for further analyses.

### 3. Results and Discussion

The pH can affect the chemical stability of drugs in bulk, as well as in their dosage forms, and, finally, their effectiveness and safety in patients. Thus, the pH of the products is often the stability-controlling factor for many drugs [19,20]. On the other hand, there is a great need to look deeply into the stability of drugs in different pH conditions, in comparison with requirements presented currently in pharmacopoeias. Such experiments are a chance to identify new impurities of drugs as well as prevent drug degradation [1].

In order to obtain quantitative results from our degradation experiments, reliable LC-UV methods were elaborated and validated according to the official guidelines [21]. The RP18, RP8 and CN columns and mobile phases of different pH, containing acetonitrile or methanol as organic modifiers, were tested to elaborate simple assays with suitable retention times and good resolution of the peaks of interest. For the KETO assay, the RP18 column with the mobile phase containing both acetonitrile and methanol as organic modifiers was the best chromatographic system. For EPI and EME, the CN column was the better choice with the mobile phases containing either only acetonitrile (for the EPI assay) or only methanol (for the EME assay). Formic acid or buffers of pH 2.5 and 4.8 were used for optimizing the chromatographic parameters. The proposed LC-UV methods were found to be specific, since they were able to separate KETO, EPI or EME from excipients present in their pharmaceuticals as well as from probable degradation products. All of them were very short which allowed for lower consumption of mobile phases and organic solvents. Representative chromatograms of KETO, EPI and EME in their ocular drops (A) and in the samples stressed in 0.1 M NaOH (B) are presented in Figure 2.

All results for system suitability and validation are summarized in Table 1. The acceptance criteria for system suitability were estimated as repeatability of peak areas and satisfactory tailing factors. The calculated RSD values for the peak areas (peak area of the drug versus peak area of the internal standard) were 0.97%, 0.84% and 0.65% (*n* = 6), while the peak tailing values were 1.88, 1.35 and 1.13 for KETO, EPI and EME, respectively. Thus, the acceptance criteria for the suitability system, i.e., RSD below 1% and the peak tailing not higher than 2, were fulfilled [22].

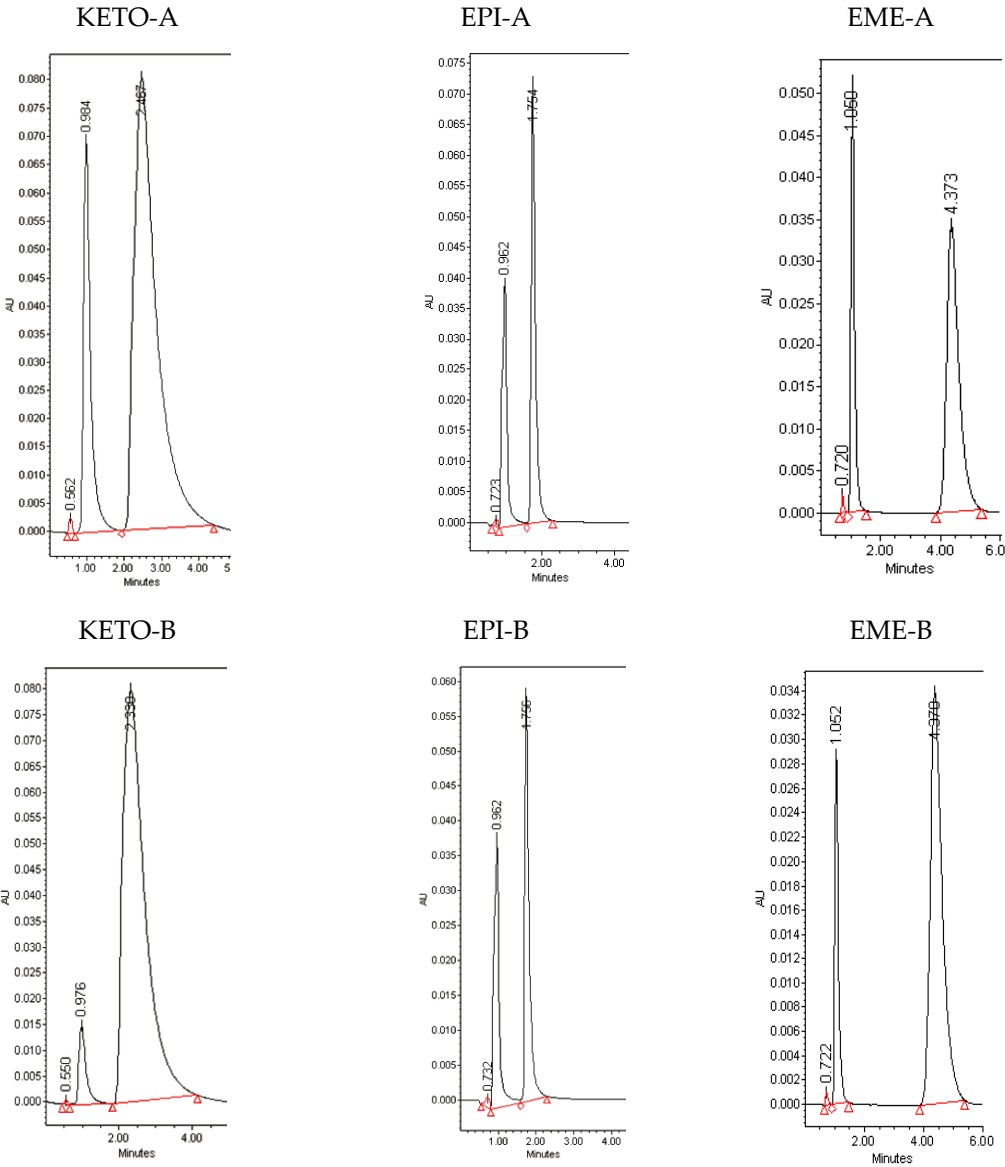

**Figure 2.** Representative chromatograms of KETO ($t_R$ = 0.984 min), EPI ($t_R$ = 0.962 min) and EME ($t_R$ = 1.052 min) in their ocular drops (**A**) and in the samples stressed in 0.1 M NaOH (**B**) (in the presence of internal standards).

**Table 1.** Validation of HPLC methods for determination of ketotifen (KETO), epinastine (EPI) and emedastine (EME) (*n* = 5).

| Parameter | Values | | |
|---|---|---|---|
| | **KETO** | **EPI** | **EME** |
| Retention time (min) | 0.98 | 0.96 | 1.05 |
| Internal standard (i.s.) | 2.47 | 1.87 | 4.07 |
| Resolution (between the drug and i.s.) | 2.11 | 1.39 | 2.86 |
| RSD for system suitability (%) | 0.97 | 0.84 | 0.65 |
| Tailing factor for the drugs | 1.88 | 1.35 | 1.13 |
| Tailing factor for i.s. | 1.94 | 1.21 | 1.08 |
| Linearity range (μg/mL) | 10–100 | 10–100 | 10–100 |
| Slope | 0.0061 | 0.03020 | 0.01993 |
| SD of the slope | 0.00002 | 0.00028 | 0.00013 |
| Intercept | 0.00181 | 0.04752 | 0.06142 |

**Table 1.** *Cont.*

| Parameter | Values | | |
|---|---|---|---|
| | **KETO** | **EPI** | **EME** |
| SD of the intercept | 0.00012 | 0.00475 | 0.00089 |
| $r^2$ | 0.9997 | 0.9962 | 0.9996 |
| SD of the $r^2$ | 0.00018 | 0.00042 | 0.00031 |
| LOD (µg/mL) | 0.06 | 0.47 | 0.15 |
| LOQ (µg/mL) | 0.21 | 1.57 | 0.45 |
| Accuracy (% Recovery) | 100.49 | 100.44 | 100.93 |
| SD of Recovery | 1.74 | 1.61 | 1.36 |
| One-day precision (% RSD) | 0.24 | 1.02 | 1.86 |
| Inter-day precision (% RSD) | 0.21 | 1.03 | 1.98 |

The calibration curves were found to be linear over the range 10–100 µg/mL for KETO, EPI and EME, with the slope RSD of 0.33%, 0.93% and 0.65% (*n* = 5), respectively. The results showed that within the chosen concentration range, there was a sufficient correlation between the peak area ratios and the concentrations of the drugs. The LOD values for KETO, EPI and EME were found as 0.06, 0.47 and 0.15 µg/mL. The LOQ values were calculated as 0.21, 1.57 and 0.45 µg/mL, respectively. While the precision of the methods was examined at three concentrations (low, medium and high) of KETO, EPI and EME, mean RSDs of 0.24%, 1.02% and 1.86% (the one-day precision, *n* = 3) and 0.21%, 1.03% and 1.98% (the inter-day precision, *n* = 6) were obtained. Recovery data obtained from the study of respective ocular drops ranged from 98.39 to 102.87%, with the mean values of RSD 1.02%, 1.61% and 1.36% for KETO, EPI and EME, respectively (*n* = 6). Therefore, the above results indicate the sufficient selectivity, linearity, sensitivity, accuracy and precision of the elaborated methods.

As far as KETO degradation was concerned, stronger correlations (higher $r^2$ values) were shown for the plots of logarithms of concentrations of non-degraded KETO than for the plots of concentrations of the non-degraded drug. Thus, the pseudo-first-order kinetics of degradation was confirmed (Figure 3).

It was interesting to observe that respective degradation rate constants for KETO varied depending on pH over the range $10^{-3}$–$10^{-4}$ min$^{-1}$ The calculated $t_{0.5}$ values varied from 50.15 (buffer of pH 3.0) and 25.08 (buffer of pH 7.0) through 10.03 (0.1 M HCl) to 4.18 and 3.86 h (buffer of pH 10.0 and 0.1 M NaOH), confirming considerable degradation of KETO in strong acidic and alkaline conditions (Table 2).

**Table 2.** Percentage degradation and kinetic parameters for ketotifen (KETO), epinastine (EPI) and emedastine (EME) in solutions of different pH at 70 °C.

| | **0.1 M HCl** | **Buffer 3.0** | **Buffer 7.0** | **Buffer 10.0** | **0.1 M NaOH** |
|---|---|---|---|---|---|
| | | | KETO | | |
| %degradation | 14.04 | 3.94 | 5.96 | 30.06 | 32.07 |
| $r^2$ | 0.9047 | 0.9834 | 0.9434 | 0.9868 | 0.9896 |
| k (min$^{-1}$) | $1.15 \times 10^{-3}$ | $2.31 \times 10^{-4}$ | $4.61 \times 10^{-4}$ | $2.76 \times 10^{-3}$ | $2.99 \times 10^{-3}$ |
| $t_{0.5}$ (h) | 10.03 | 50.15 | 25.08 | 4.18 | 3.86 |
| | | | EPI | | |
| %degradation | 10.83 | 9.41 | 6.22 | 5.46 | 13.76 |
| $r^2$ | 0.9671 | 0.9569 | 0.9447 | 0.9484 | 0.9042 |
| k (min$^{-1}$) | $6.91 \times 10^{-4}$ | $6.91 \times 10^{-4}$ | $4.61 \times 10^{-4}$ | $4.61 \times 10^{-4}$ | $9.21 \times 10^{-4}$ |
| $t_{0.5}$ (h) | 16.72 | 16.72 | 25.08 | 25.08 | 12.54 |
| | | | EME | | |
| %degradation | 35.06 | 29.26 | 39.86 | 38.72 | 51.88 |
| $r^2$ | 0.9764 | 0.9807 | 0.9950 | 0.9644 | 0.9734 |
| k (min$^{-1}$) | $3.22 \times 10^{-3}$ | $2.53 \times 10^{-3}$ | $3.92 \times 10^{-3}$ | $3.92 \times 10^{-3}$ | $5.53 \times 10^{-3}$ |
| $t_{0.5}$ (h) | 3.58 | 4.57 | 2.95 | 2.95 | 2.08 |

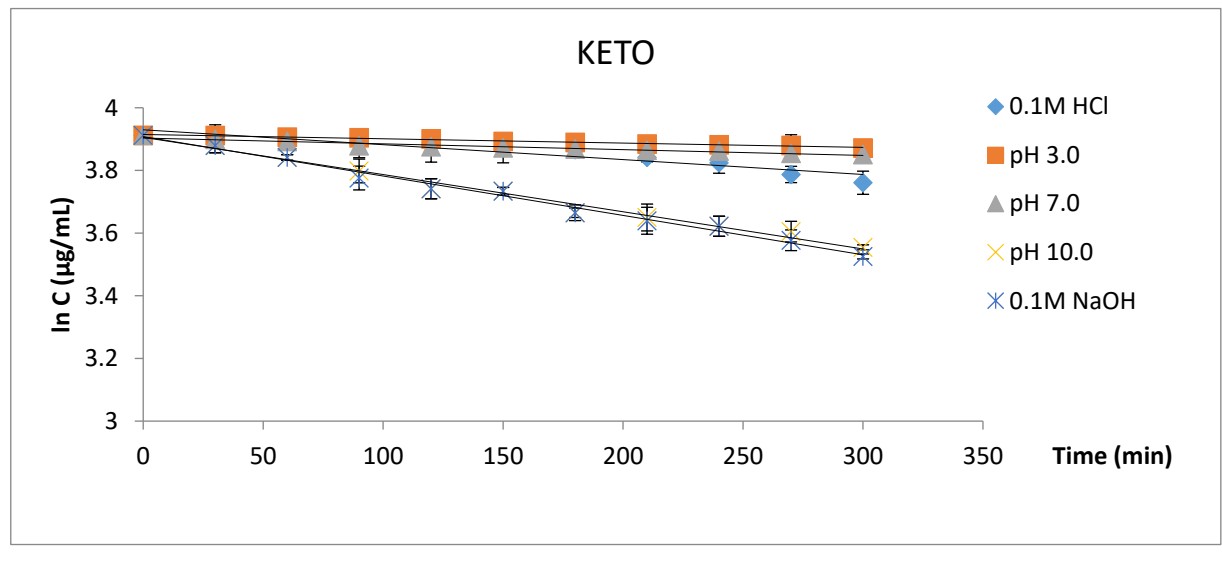

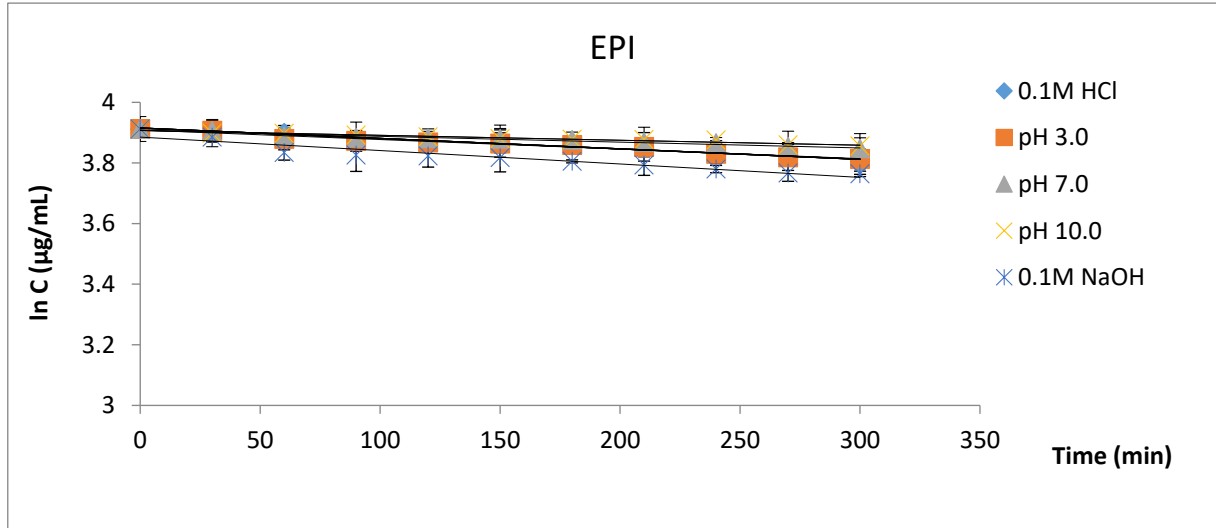

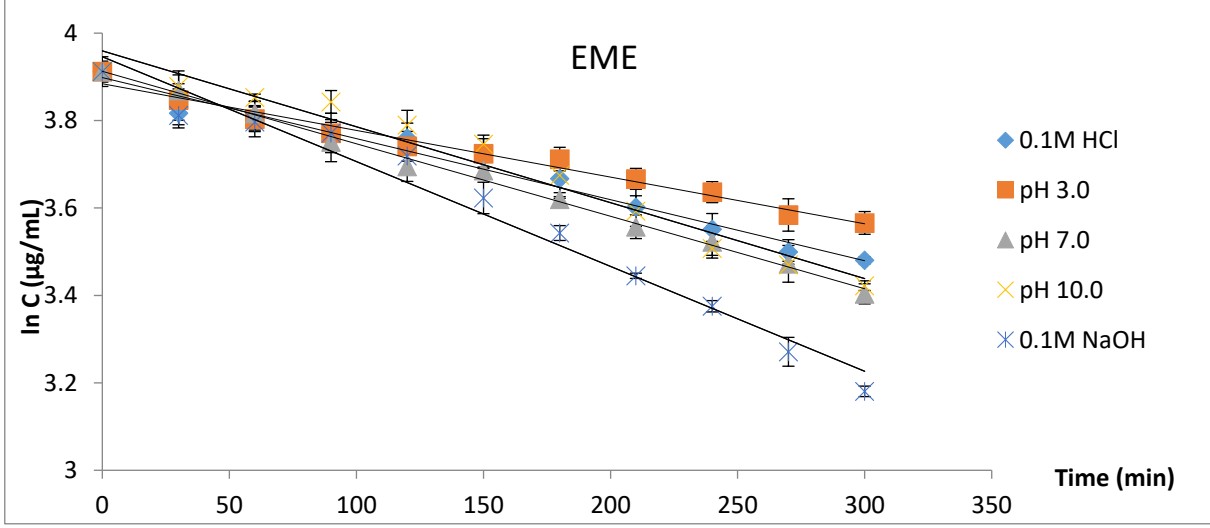

**Figure 3.** Pseudo-first-order plots for degradation of ketotifene (KETO), epinastine (EPI) and emedastine (EME) in solutions of different pH at 70 °C.

As shown above, very limited information on the stability of KETO has been published so far. In the study of Kabra et al. [6], higher sensitivity of KETO to alkaline than to acidic conditions was shown. Similarly, in the study of Elsayed [7], degradation of KETO in basic conditions resulted in significant decomposition, and the final content of the parent drug was only 3.6%. This suggests poor stability of KETO under strong basic conditions which was visibly supported by the present results. However, the sensitivity of KETO to strong acidic conditions should also be considered, especially since the previous experiments from our laboratory showed high photodegradation of KETO at pH 3.0 [18]. On the other hand, we can now summarize the results from both experiments and conclude that KETO is visibly more sensitive to light than to high temperature under the same pH conditions.

Our UPLC-MS/MS experiments led to obtain the full scan mass spectra of KETO in which the protonated molecule [M + H]$^+$ of *m/z* 310.1 was observed. Then, its MS/MS showed the product ion at *m/z* 249.0 that formed a subsequent ion at *m/z* 221.0. A protonated KETO molecule also followed other fragmentation pathways, the first one forming an ion at *m/z* 213.1, with the subsequent ions at *m/z* 199.0 and *m/z* 185, and the second one forming an ion at *m/z* 96.1 (Table 3).

**Table 3.** MS/MS results for ketotifen (KETO) and its degradation products (K-DPs) at 70 °C.

| Compound | $t_R$ (min) | [M + H] $^+$ | Fragmentation Ions | Structures |
|---|---|---|---|---|
| KETO | 3.79 | 310.1 | 249.0, 221.0, 213.1, 199.0, 185.0, 96.1 |  |
| K-DP1 0.1 M HCl Buffer 3.0 Buffer 7.0 Buffer 10.0 | 3.43 | 342.1 | 324.1, 306.1, 278.1, 221.0 |  |
| K-DP2 0.1 M HCl Buffer 3.0 Buffer 7.0 Buffer 10.0 | 3.69 | 342.1 | 324.1, 306.1, 278.1, 250.1, 221.0 |  |
| K-DP3 0.1 M HCl Buffer 10.0 0.1 M NaOH | 3.46 | 358.1 | 322.1, 276.0, 264.0, 248.1, 236.1, 221.0 |  |

**Table 3.** *Cont.*

| Compound | $t_R$ (min) | $[M + H]^+$ | Fragmentation Ions | Structures |
|---|---|---|---|---|
| K-DP4 Buffer 7.0 Buffer 10.0 0.1 M NaOH | 3.55 | 374.1 | 322.1 | |
| K-DP5 Buffer 7.0 0.1 M NaOH | 3.56 | 320.1 | 292.1, 275.1, 263.1, 247.1 | |

Most of the stress conditions used in the present study led to obtain two degradation products of KETO at *m/z* 342.1, i.e., K-DP1 with $t_R$ = 3.43 min and K-DP2 with $t_R$ = 3.69 min. After stressing with 1 M HCl, buffer of pH 10 and 0.1 M NaOH, the next product was detected at $t_R$ = 3.46 min, i.e., K-DP3 at *m/z* 358.1, with the subsequent fragment ions at *m/z* 3221.1, 276.0, 264.0, 248.1, 236.1 and 322.1. In addition, the product K-DP4 at $t_R$ = 3.55 min and *m/z* 374.1 was detected when the buffer of pH 7.0 and 0.1 M NaOH were used. All above degradation products seemed to be the results of oxidation or oxidation and demethylation affecting the piperidine ring in the KETO molecule. In addition, the possibility of oxidation of the methyl group at the nitrogen atom of piperidine to the formyl group, leading to the product K-DP3 ($t_R$ = 3.56 min and *m/z* 320.1), was found in the samples stressed at pH 7.0 and 13.0 (0.1 M NaOH) (Table 3). The UPLC chromatogram showing all above degradation products of KETO is presented in Figure 4.

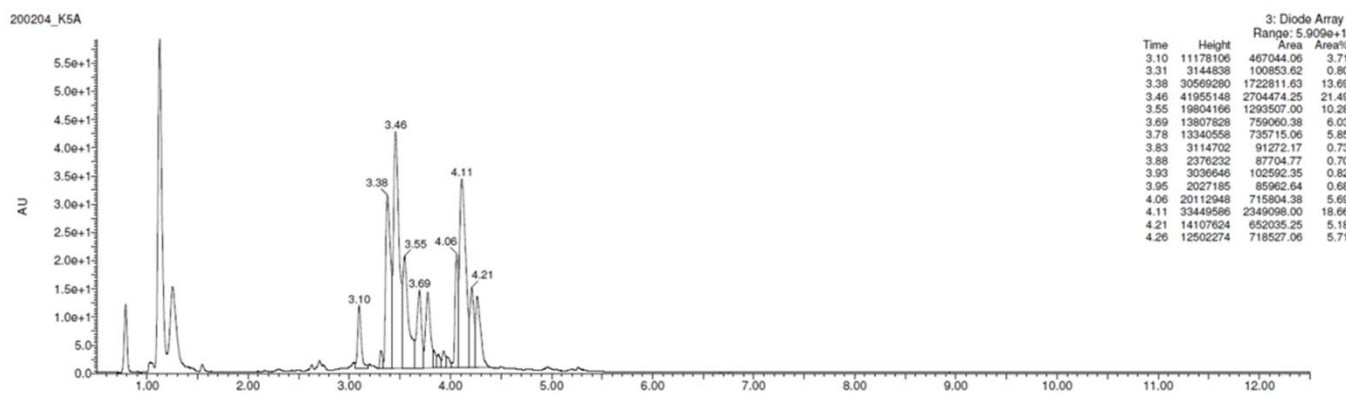

**Figure 4.** UPLC chromatogram of ketotifen (KETO) and its degradation products in buffer of pH 10.0 at 70 °C.

It is worth mentioning that the products K-DP3 and K-DP4 have not been reported so far. In turn, K-DP1, K-DP2 and K-DP5 were previously detected after degradation of KETO under UV/VIS light over the pH range of 3–10 [18]. Taking all this into account, we can assume that decomposition of KETO at a specific pH under high temperature could follow similar mechanisms to those occurring under UV/VIS light. However, it should be remembered that the light intensified most of the above decomposition processes. On the other hand, it should be emphasized that two degradation products, i.e., K-DP3 and K-DP4, were detected only in the present experiment after degradation at high temperature and

not detected under UV/VIS irradiation. Hence, their fragmentation patterns are shown in Figures 5 and 6. In turn, a comprehensive scheme for KETO degradation in solutions of different pH is proposed in Figure 7.

**Figure 5.** Fragmentation pattern of ketotifen (KETO) degradation product K-DP3.

**Figure 6.** Fragmentation pattern of ketotifen (KETO) degradation product K-DP4.

**Figure 7.** Proposed degradation pathways of ketotifen (KETO) in different pH conditions at 70 °C.

The data from the literature concerning KETO [23] suggest that similar products could be formed in the body during metabolic processes. The drug is mainly metabolized in the liver and the main metabolites were confirmed as KETO-*N*-glucuronide, nor-KETO and the 10-hydroxy compound. Thus, at least in part, the metabolism of KETO occurs through the proposed degradation pathways. What is more, *N*-demethylated degradants of KETO, i.e., K-DP2, K-DP3 and K-DP4, were identified after degradation in a wide pH range from 1 to 13.

As far as EPI is concerned, its degradation could be described by the pseudo-first-order kinetics (Figure 3) with all rate constants at the level of $10^{-4}$ min$^{-1}$, and with percentage degradation below 14%. The calculated $t_{0.5}$ values were 16.72 h (acidic conditions), 25.08 h (pH 7–10) and 12.54 h (0.1 M NaOH), indicating moderate sensitivity of EPI to strong acidic and strong alkaline conditions (Table 2). In some previous studies from the literature [8–10], EPI was stressed under acidic, neutral and alkaline conditions, as a part of validation of quantitative HPLC methods. In two studies, the sensitivity of EPI to alkaline conditions was observed, whereas its stability in acidic and neutral conditions was proved [11,12]. On the other hand, the much higher susceptibility of EPI to acidic than alkaline conditions was reported in another experiment [13]. The present results strongly suggest sensitivity of EPI to acidic as well as alkaline conditions.

When our UPLC-MS/MS analysis was performed, EPI showed a protonated molecule, [M + H]$^+$, at *m/z* 310.1, while its MS/MS showed the product ions at *m/z* 249.0, 213.1 and 96.1. One degradation product, i.e., EPI-DP1 with *m/z* 268.1 (250.1, 225.1, 208.1, 193.1, 130.1), was detected in the samples stressed in strong alkaline conditions (buffer of pH 10.0 and 0.1 M NaOH). The respective UPLC chromatogram showing the product EPI-DP1 with $t_R$ = 3.06 min is presented in Figure 8.

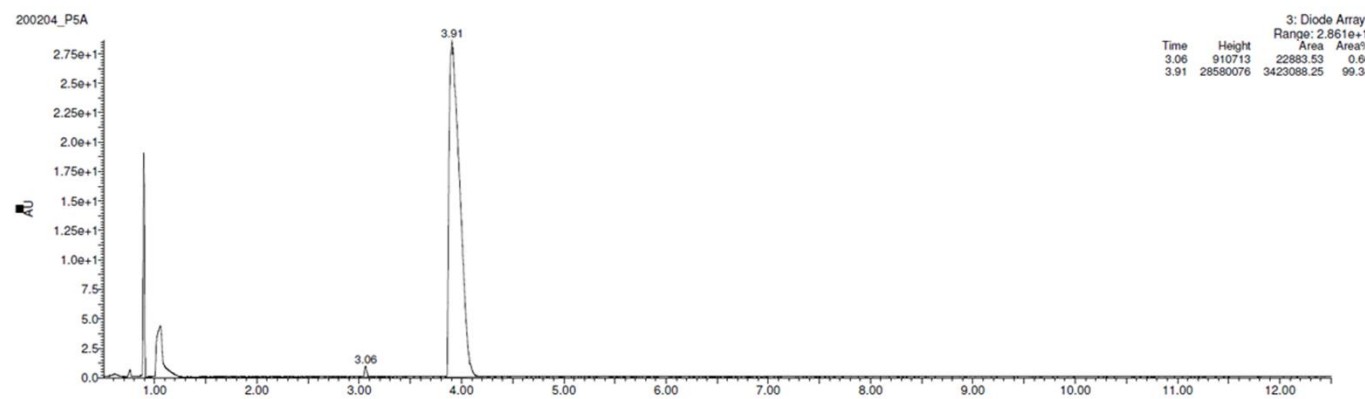

**Figure 8.** UPLC chromatogram of epinastine (EPI) and its degradation product in 0.1 M NaOH at 70 °C.

As a result, alkaline degradation through opening of the imidazole ring in the EPI molecule with formation of the amide group was proposed (Table 4). The fragmentation pattern of the product EPI-DP1 is shown in Figure 9.

**Table 4.** MS/MS results for epinastine (EPI) and its degradation product EPI-DP1 at 70 °C.

| Compound | $t_R$ (min) | [M + H]$^+$ | Fragmentation Ions | Structures |
|---|---|---|---|---|
| EPI | 3.79 | 250.1 | 208.1, 193.1, 178.1, 165.1, 130.1, 115.1, 91.1, | |
| EPI-DP1 Buffer pH 10.0 0.1 M NaOH | 3.06 | 268.1 | 250.1, 225.1, 208.1, 193.1, 130.1 | |

**Figure 9.** Fragmentation pattern of epinastine (EPI) degradation product EPI-DP1.

It is interesting to conclude that the EPI-DP1 compound was detected after alkaline degradation at high temperature and not detected under UV/VIS irradiation in any pH

conditions, which suggested different degradation mechanisms. The proposed degradation pathway for EPI in alkaline conditions at high temperature is shown in Figure 10.

**Figure 10.** Proposed degradation pathway of epinastine (EPI) in alkaline conditions at 70 °C.

As far as EME was concerned, its degradation at different pH and high temperature followed the pseudo-first-order kinetics, too (Figure 3). The drug was shown to be easily degraded and similarly sensitive to low and high pH values, with all rate constants at the level of $10^{-3}$ min$^{-1}$. The calculated $t_{0.5}$ values were 3.58 h and 4.57 h (0.1 M HCl and buffer of pH 3.0), 2.95 h (pH 7.0 and 10.0) and 2.08 h (0.1 M NaOH) (Table 2). At the same time, EME was shown to be much more sensitive to high temperature than to UV/VIS irradiation at similar pH conditions [18]. When UPLC-MS/MS analysis was performed, EME showed a protonated molecule, [M + H]$^+$, at *m/z* 303.2, while its MS/MS showed the product ions at *m/z* 246.2, 232.1, 218.1, 200.1, 174.1, 146.1 and 134.1 (Table 5). However, no degradation product was detected in the samples stressed at high temperature. On the contrary, two degradation products of EME were detected and identified after photodegradation, when either the ethoxy moiety or 1,4-diazepine ring in the EME molecule were affected [18]. A literature survey revealed one more study in which acidic degradation of EME was described. One degradation product was isolated using the TLC method and identified as a dimmer of 1-(2-ethoxyethyl)-2,3-dihydro-1*H*-benzimidazole with the molar mass equal to 380 [15].

**Table 5.** MS/MS results for emedastine (EME).

| Compound | $t_R$ [min] | [M + H]$^+$ | Fragmentation Ions | Structure |
|---|---|---|---|---|
| EME | 2.20 | 303.2 | 246.2, 232.1, 218.1, 200.1, 174.1, 146.1, 134.1 |  |

"Degradation rate constants of drug substances are generally affected by pH because most degradation pathways are catalyzed by hydronium and/or hydroxide ions. Water itself is also a critical reactant" [24]. KETO, EPI and EME are all weak bases with pKa values in the range 8.43–8.77 [23] because of the substituted piperidine ring, primary amine group in imidazo[1,5-a]azepine and substituted diazepine ring present in their structures, respectively. However, they differ as far as their degradation in a specific pH is concerned. The influence of pH on the reaction rate constants for KETO, EPI and EME is depicted in Figure 11.

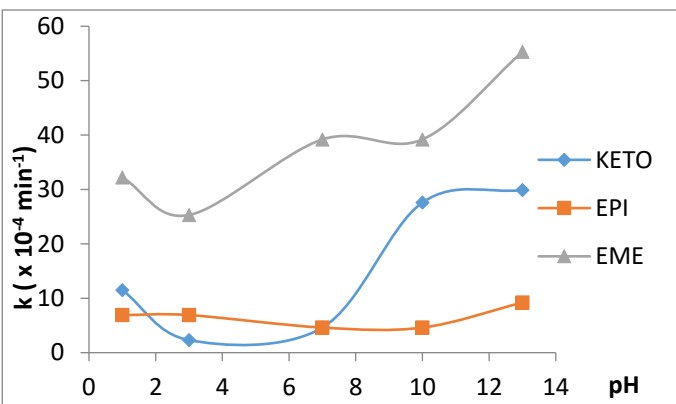

**Figure 11.** pH rate profiles for degradation of ketotifen (KETO), epinastine (EPI) and emedastine (EME) at 70 °C.

We could observe relatively greater stability of KETO in slightly acidic or neutral pH between 3 and 7. On the other hand, the degradation rate constants for KETO increased when the pH of the solution was low (<3) or high ($\geq$10). EPI was similarly sensitive to degradation over a whole pH range with percentage degradation below 14%. Their major reactions occurred at pH equal to or above 10 and were shown to affect the piperidine or the imidazole rings of KETO or EPI, respectively. Thus, the basic amine functions in heterocyclic structures of KETO and EPI that are presumed to be essential when the drugs bind to the $H_1$ receptor could be affected during degradation. The susceptibility of these structural amino functions to chemical changes may therefore increase the risk of degradation of these drugs during manufacturing of appropriate formulations, as well as the risk of their incomplete therapeutic efficacy in patients.

As far as EME was concerned, it was similarly sensitive to degradation over a whole pH range but its percentage degradation was much higher than EPI (all above 30%). At the same time, the shape of its pH rate profile was more complex with the shoulder between pH 6 and 10, reflecting changes in the mechanisms of degradation with changes in pH values. Designing further experiments allowing the detection and identification of specific degradation products of EME would therefore be another analytical challenge.

## 4. Conclusions

In the present study, the appropriate LC-UV methods were developed and validated to investigate the stability of three important $H_1$ antihistaminic drugs, i.e., KETO, EPI and EME, in a wide pH range. With these methods, KETO was found relatively stable in moderate acidic and neutral conditions and sensitive to acidic degradation. EPI was shown as moderately stable over a wide pH range, while EME was highly sensitive to all pH conditions. We identified new degradation products of KETO and EPI and proposed possible degradation pathways. The data obtained here contribute to a better understanding of the stability characteristics of KETO, EPI and EME. The presented results could also be helpful in developing new $H_1$ receptor antagonists with higher activity, lower side effects and higher chemical stability. The results presented here supplement the literary resources and could be important for the drugs' manufacturing and storage conditions and, finally, for their safety in patients.

**Author Contributions:** Conceptualization, A.G.; methodology, A.G., P.Ż. and I.K.; software, A.G., U.H. and P.Ż.; validation, A.G. and I.K.; formal analysis, A.G. and U.H.; investigation, A.G., I.K. and P.Ż.; resources, A.G. and U.H.; data curation, A.G., P.Ż. and I.K.; writing—original draft preparation, A.G., U.H. and P.Ż.; writing—review and editing, A.G., U.H. and P.Ż.; supervision, U.H. All the authors approved the final manuscript for publication. All authors have read and agreed to the published version of the manuscript.

**Funding:** This research received no external funding.

**Institutional Review Board Statement:** Not applicable.

**Informed Consent Statement:** Not applicable.

**Data Availability Statement:** No new data were created or analyzed in this study. Data sharing is not applicable to this article.

**Conflicts of Interest:** The authors declare no conflict of interest. The funders had no role in the design of the study; in the collection, analyses, or interpretation of data; in the writing of the manuscript, or in the decision to publish the results.

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
