# Peer review of "LC-UV and UPLC-MS/MS Methods for Analytical Study on Degradation of Three Antihistaminic Drugs, Ketotifen, Epinastine and Emedastine: Percentage Degradation, Degradation Kinetics and Degradation Pathways at Different pH"

_processes, doi:10.3390/pr9010064_

Round 1
Reviewer 1 Report
This is a well-written and well-referenced article, some minor issues are needed:
Abstract section:
-at line 21, ....of various pH (at five different values 1,3,7,10,13)... not ...(1-13)...;
-at line 25, ....at pH 1-7 (degradation ≤ 14.04 %) ... not ...(≥ 14.04 %)...;
-at line 31, I think that it should be added...any degradation products for emedastine were not detected....
Author Response
Abstract section:
-at line 21, ....of various pH (at five different values 1,3,7,10,13)... not ...(1-13)...;
It was corrected.
-at line 25, ....at pH 1-7 (degradation ≤ 14.04 %) ... not ...(≥ 14.04 %)...;
Thank you. It was corrected.
-at line 31, I think that it should be added...any degradation products for emedastine were not detected....
It was added.
Reviewer 2 Report
-It'll be interesting to add the structures of ketotifen, epinastine and emedastine at the introduction (although the structures of two of them, ketotifen and epinastine, could be found throughout the text, the third is not available).
-Compounds KH2PO4 and KOH (line 96) are usually called potassium dihydrogen phosphate and potassium hydroxide instead of kalium dihydrogen phosphate and kalium hydroxide.
-The structures of table 4 seem to be missing (at least in my pdf).
Author Response
-It'll be interesting to add the structures of ketotifen, epinastine and emedastine at the introduction (although the structures of two of them, ketotifen and epinastine, could be found throughout the text, the third is not available).
The structures of all drugs were added as a new Figure 1. Additionally, Table 5 presenting the protonated molecule of emedastine was added.
-Compounds KH2PO4 and KOH (line 96) are usually called potassium dihydrogen phosphate and potassium hydroxide instead of kalium dihydrogen phosphate and kalium hydroxide.
This part was extensively shorted according to the Editor suggestions. Thus, the names for some reagents are not specified now.
-The structures of Table 4 seem to be missing (at least in my pdf).
It was checked.
Reviewer 3 Report
In the present paper stability of three antihistaminic drugs named ketotifen, epinastine, and emedastine was investigated within a large range of pH and high temperature using LC-UV method.
New degradation products of of ketotifen, epinastine were identified and possible degradation pathways were proposed using the UPLC-MS/MS method. All the results were compared with previous developed studies and the new developed findings can be important for manufacturing and storage conditions of these drugs as well as for evaluate their safety in patients.
The paper is well written, the experimental part is well presented, the results are correctly interpreted and compared with literature data previously obtained.
However, I believe that the presentation of some aspects related to the metabolism processes of the studied drugs and the selection of the chosen experimental conditions would improve the manuscript. For this, a short discussion of the following aspects would be added in order to improve the manuscript:
- What is the general route of metabolization of these three drugs? More precisely in what pH conditions are they subjected during metabolization processes? It is possible that degradation compounds could form in the body during metabolic processes? These aspects would be important regarding the safety in patients.
- Why the temperature of 70oC was chosen to investigate the degradation processes? Was this temperature chosen according to studies developed in the literature? Is it possible for these degradation compounds to form at normal storage temperatures (for example 220C-370C)?
- A discussion / correlation with the possibility of the formation of degradation compounds in the body would be useful in order to correlate with the known possible adverse reactions of these drugs?
Author Response
However, I believe that the presentation of some aspects related to the metabolism processes of the studied drugs and the selection of the chosen experimental conditions would improve the manuscript. For this, a short discussion of the following aspects would be added in order to improve the manuscript:
- What is the general route of metabolization of these three drugs? More precisely in what pH conditions are they subjected during metabolization processes? It is possible that degradation compounds could form in the body during metabolic processes? These aspects would be important regarding the safety in patients.
Thank you very much for this interesting suggestion.
Our results for KETO and the literature data suggest that degradation of the drug could be similar, at least in part, to metabolic processes in the body. The drug is mainly metabolized in the liver and three main metabolites are described in the literature as KETO-N-glucuronide, nor-KETO and the 10-hydroxy metabolite. What is more, N-demethylated degradants of KETO, i.e. K-DP2, K-DP3 and K-DP4 were identified after degradation in a wide pH range from 1 to 13. As far as EPI is concerned, its metabolism occurs mainly in the liver, but the degree of metabolism is reported to be very low. In the case of EME, it was shown to be converted by oxidative biotransformation to metabolites, 5-hydroxyemedastine and 6-hydroxyemedastine. Minor metabolites include the 5'-oxoanalogs of 5-hydroxyemedastine and 6-hydroxy-emedastine and the N-oxide. As was shown above, any similar degradation products for EME were not detected in our stressed samples. The short discussion on the above questions was added in our revised text (lines 345-350).
- Why the temperature of 70oC was chosen to investigate the degradation processes? Was this temperature chosen according to studies developed in the literature? Is it possible for these degradation compounds to form at normal storage temperatures (for example 220C-370C)?
Our experiments were started at room temperature. However, degradation of the examined drugs was too slow. The main goal of the present study was to obtain as many as possible degradation products, thus the temperature of 70°C was applied which was acceptable for the stress degradation conditions.
- A discussion / correlation with the possibility of the formation of degradation compounds in the body would be useful in order to correlate with the known possible adverse reactions of these drugs?
Thank you very much for this interesting suggestion.
On the one hand, the second-generation H1-antihistamines are described as relatively free of significant adverse effects. On the other hand, the increased risk of cardiac adverse effects for some H1-antihistamine receptor antagonists that prolong the QT interval was shown. In addition, some of them are reported to cause photosensitivity, urticaria, fever, hepatitis and others. Unfortunately, the mechanisms for these adverse effects are not fully understood yet. The presented results would be certainly useful for further study on possible adverse effects of H1 antihistaminic antagonists due to their metabolites and/or degradants.